# Adiabatic Shear Banding in Nickel and Nickel-Based Superalloys: A Review

**Russell A. Rowe [1], Paul G. Allison [2], Anthony N. Palazotto [3] and Keivan Davami [1,\*]**

1    Department of Mechanical Engineering, University of Alabama, Tuscaloosa, AL 35487, USA
2    Department of Mechanical Engineering, School of Engineering and Computer Science, Baylor University, Waco, TX 76798, USA
3    Department of Aeronautics and Astronautics, Air Force Institute of Technology, Wright-Patterson AFB, Dayton, OH 45433, USA
\*    Correspondence: kdavami@eng.ua.edu

**Abstract:** This review paper discusses the formation and propagation of adiabatic shear bands in nickel-based superalloys. The formation of adiabatic shear bands (ASBs) is a unique dynamic phenomenon that typically precedes catastrophic, unpredicted failure in many metals under impact or ballistic loading. ASBs are thin regions that undergo substantial plastic shear strain and material softening due to the thermo-mechanical instability induced by the competitive work hardening and thermal softening processes. Dynamic recrystallization of the material's microstructure in the shear region can occur and encourages shear localization and the formation of ASBs. Phase transformations are also often seen in ASBs of ferrous metals due to the elevated temperatures reached in the narrow shear region. ASBs ultimately lead to the local degradation of material properties within a narrow band wherein micro-voids can more easily nucleate and grow compared to the surrounding material. As the micro-voids grow, they will eventually coalesce leading to crack formation and eventual fracture. For elevated temperature applications, such as in the aerospace industry, nickel-based superalloys are used due to their high strength. Understanding the formation conditions of ASBs in nickel-based superalloys is also beneficial in extending the life of machining tools. The main goal of the review is to identify the formation mechanisms of ASBs, the microstructural evolutions associated with ASBs in nickel-based alloys, and their consequent effect on material properties. Under a shear strain rate of 80,000 s$^{-1}$, the critical shear strain at which an ASB forms is between 2.2 and 3.2 for aged Inconel 718 and 4.5 for solution-treated Inconel 718. Shear band widths are reported to range between 2 and 65 microns for nickel-based superalloys. The shear bands widths are narrower in samples that are aged compared to samples in the annealed or solution treated condition.

**Keywords:** split Hopkinson pressure bar; dynamic shear localization; adiabatic shear band; microhardness testing

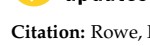


## 1. Introduction

ASBs form in many metals at high strain rates and large plastic deformation due to the localization of plastic flow into a concentrated region. This localization occurs mainly at high strain rates once a critical shear strain value is reached and can produce large amounts of heat as approximately 90% of the work of deformation is converted into thermal energy [1]. At high rates of deformation, this heat may not have sufficient time to dissipate away from the shear region leading to thermal softening. Since the region is plastically deformed, it also undergoes work hardening and thus this region of adiabatic shear is under a plastic instability phenomenon. Predicting at what stage an ASB will form is difficult, but a general rule is that an ASB can form when the strength loss due to thermal softening exceeds the strength gained from work or strain rate hardening [2,3]. This naturally means that materials with low strain/strain rate hardening coefficients and/or

low thermal conductivity have an increased risk of ASB formation. ASBs commonly occur in aluminum, titanium, uranium alloys, and steels during ballistic impact [4,5], however, nickel-based alloys have a higher resistance to localized shear band formation [2]. The initiation of shear bands can be furthered by the presence of microstructural defects or voids as these artifacts can further localize stresses in the shear region [6]. Understanding the formation mechanisms for ASBs can aid in the prediction of material behavior under high-speed deformation specifical when loaded under torsion or compression, during dynamic impact, high-speed machining, and explosive fragmentation.

ASBs have been classified by Rogers [7] into two distinct categories: "transformation bands" and "deformation bands". "Transformation bands" are bands in which there is a crystallographic phase change or change in the microstructural orientation as a result of the generated heat, plastic deformation, and/or rapid cooling. "Deformation bands" experience no phase change and are purely plastically deformed bands. ASBs occur in various materials differently and are sensitive to multiple mechanical variables such as cutting tool/workpiece geometry, rate of deformation, preheating, and temperature as well as material properties such as strain-rate sensitivity, the temperature dependence of flow stress, strain-hardening rate, thermal conductivity, specific heat, and phase transformation kinetics [8–11]. Depending on the metal used and the type of deformation, temperatures in the ASBs could be several hundred degrees higher than the surrounding material, with cooling rates on the order of $10^7$ K/s [12,13].

ASBs form intermittently throughout the shear region and eventually coalesce to form a singular band [14]. Material that has been additively manufactured (AM) [15] commonly contains imperfections, such as microvoids, which create stress concentrations and allow ASBs to form at lower strains compared to traditionally manufactured (TM) material [16]. In TM materials, this occurs in a straight line, primarily along the plane of maximum shear stress. However, in AM materials, microvoids may be irregularly distributed in the shear region and therefore cause ASBs to form in an irregular path. At a cutting speed of 2 m/s, Inconel 625 that was manufactured through casting did not show evidence of ASB formation (Figure 1a), however, AM Inconel 625 built using selective laser melting did show distinct ASBs (Figure 1b red arrow) under the same cutting conditions. As the cutting speed increased to 4 m/s, ASBs began to arise in the cast material, forming along a straight path in contrast to the shear bands formed in the AM samples which took an irregular curved path. Bhavsar et al. stated that the irregular path the ASB takes supported their hypothesis that the shear band formation in the AM samples occurs near microvoids inherent in the selective laser melting (SLM) process [16].

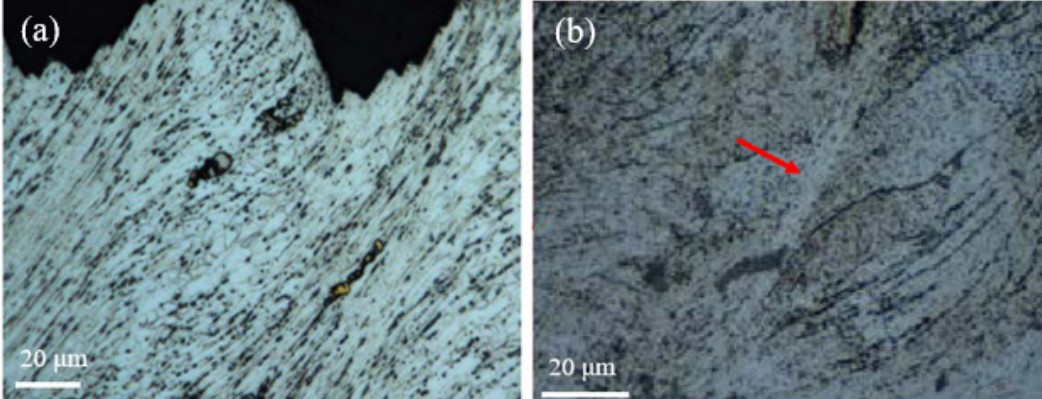

**Figure 1.** Under the same post-processing cutting operation, (**a**) no visible ASB can be seen in cast IN 625, (**b**) however distinct ASB formation is observed in SLM IN 625 showing that AM material is more susceptible to ASB formation [16]. Reprinted from [16] with permission from Elsevier.

ASB formation leads to a degradation of mechanical properties by weakening the material in the shear band thus creating a pathway for cracks to preferentially travel

along [14]. Identification of a shear band optically in an etched material is relatively simple because the boundary between the band and the surrounding material is very distinct and can be distinguished from other types of shear failure in Nickel-based superalloys by the presence of recrystallized grains in the shear band. One can estimate the width of a shear band analytically using the half-width model proposed by Dodd and Bai [17], where $\delta$ is the predicted half-width of the shear band, $k$ is the materials thermal conductivity, $T$ is half of the melting temperature, $\tau$ is the shear stress and $\dot{\gamma}$ is the shear strain rate:

$$\delta = \left( \frac{kT}{\tau \dot{\gamma}} \right)^{\frac{1}{2}} \tag{1}$$

Understanding what factors increase the likelihood of ASB formation in metals, such as heat treatments [18,19] or elevated temperature [10], is important as this information can be used to optimize cutting parameters for high-speed machining [20–22]. In this literature review, the following will be discussed for Nickel-based superalloys: shear band formation and propagation kinetics, and how the microstructures around the ASBs evolve when shear localization occurs.

## 2. Testing Methodology

Numerous testing methods have been developed to characterize the formation of ASBs and further understand the effect of shear localization in a material. The most common are torsional testing [23–26] (Figure 2a), forced shear compression testing [25,27–29] (Figure 2b,c), and high-speed cutting with a quick stop device [30] (Figure 2d). The reasoning behind using these testing methods is that the geometries and loading conditions encourage shear deformation in a localized region. The compact forced simple shear (CFSS) sample geometry is the most recent design for testing materials under simple shear [28,31]. This design is an improvement upon the previously used top hat geometry which had significant issues such as rotation of the shear surface during testing and radial expansion of the brim causing a multi-axial stress state [28]. The CFSS sample design allows a narrow plane to be under simple shear loads, shown by the arrows in Figure 2b, as the two ends are pushed together with no compressive loading in the shear section. One significant advantage that the top-hat geometry does have over the CFSS is that stopper rings may be used with the top-hat samples to limit their deformation which allows for characterizing the formation process of ASBs more accurately [32].

High-speed cutting experiments are primarily used to determine the optimal cutting speed to prevent serrated chip formation, which is merely repeated ASB formation, as their formation increases the wear on cutting tools [30,33]. Cutting experiments are typically done using a quick-stop device to disengage the cutting tool and "freeze" the cutting state for analysis. The repeated formation of ASBs causes micro-cracks on the cutting tool increasing the wear rate and lowing the tool's lifetime. Factors such as workpiece feed rate and the cutting angle can affect the stresses on the workpiece however cutting speed has the most obvious influence on shear band formation [8,9].

During a compressive test on a top hat sample, Figure 2c, the tip of the sample will be pushed into the hollow brim section of the sample creating a shear region shown in red. During this test the radial expansion of the brim is limited and thus most of the strain measurement is from shearing. The shear strain achieved can be limited by using a stopper ring made of stronger material, preferably the same material as the incident/transmitted bars. The compressive and torsional Hopkinson bars are shown in Figure 3a,b, respectively. Assuming that the specimen is in equilibrium, the stress $\tau_s$, strain $\gamma_s$, and strain rate $\dot{\gamma}_s(t)$ can be evaluated using the following equations listed for a top-hat sample [32,34].

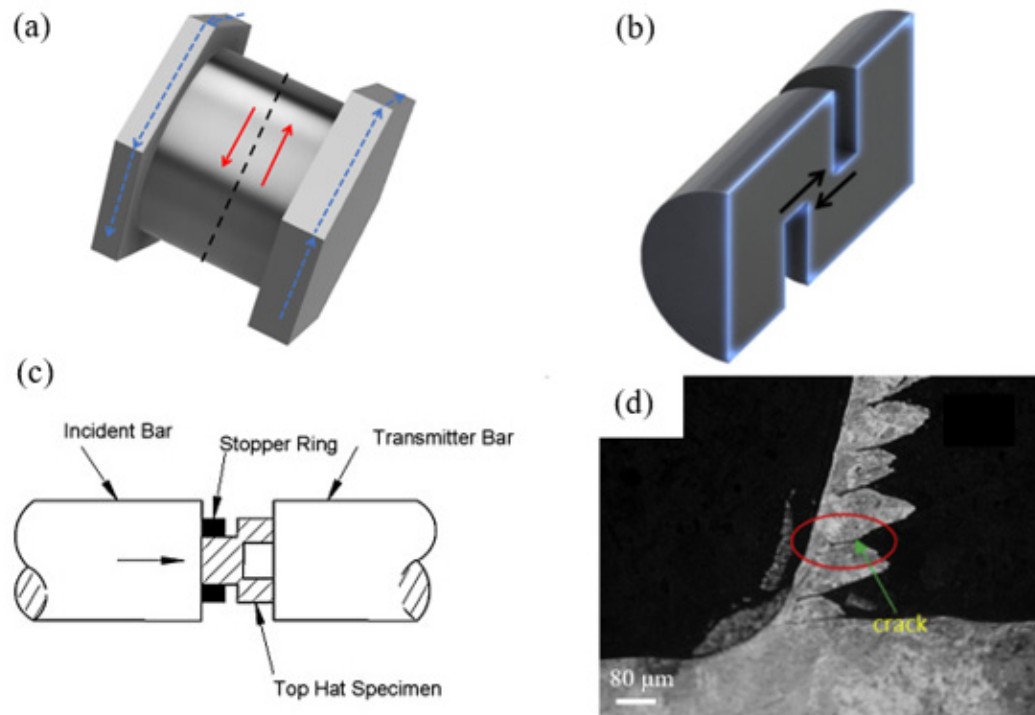

**Figure 2.** (**a**) Torsional loading, (**b**) CFSS sample, (**c**) top-hat forced shear, and (**d**) high-speed cutting for studying shear band formation [5,27,30]. The scale bar in (**d**) is 80 microns. Reprinted from [5,27,30] with permission from Elsevier.

$$\tau_s(t) = \frac{E_{bar} A_{bar}}{A_{specimen}} \varepsilon_T(t) \cos(\theta) \tag{2}$$

$$\gamma_s(t) = -2 \frac{C_{bar}}{L_s} \int_0^t \varepsilon_R(t) \partial t \tag{3}$$

$$\dot{\gamma}_s(t) = -2 \frac{C_{bar}}{L_s} (\varepsilon_R(t)) \tag{4}$$

Here, $A_{bar}$ and $A_{specimen}$ are the cross-sectional area of the bar and the specimen, respectively; $E_{bar}$ is the elastic modulus of the incident and transmission bar material; $C_{bar}$ is the wave speed of the bars; $L_s$ is the length of the shear region; $\varepsilon_R(t)$ is the strain of the reflected wave in the incident bar at time $(t)$, and $\varepsilon_T(t)$ is the strain of the transmitted wave in the transmitted bar at time $(t)$.

The overall shear stress $\tau_s$ and shear strain $\gamma_s$ from tests with the torsional bars can be determined using the following equations [35,36]:

$$\tau_s = \frac{2T}{\pi D_s^2 t_s} \tag{5}$$

$$\gamma_s(t) = \left( \frac{2c_s D_s}{L_s D} \right) \int_0^t \dot{\gamma}_R(t) dt \tag{6}$$

$$\dot{\gamma}_s(t) = \left( \frac{2c_s D_s}{L_s D} \right) \gamma_R(t) \tag{7}$$

In the previous equation, $T$ is applied torque during the test; $D_s$ and $D$ are the mean diameter of the torsional sample and diameter of the bar, respectively; $t_s$ is the thickness of the test section; $\dot{\gamma}_R$ and $\gamma_R$ is the shear strain rate and shear strain, respectively, from the reflected wave; $L_s$ is the width of the gauge section; $c_s$ is the torsional wave velocity

in the bar. During torsional testing, the test specimen is twisted around the central axis of the torsional bars. The torque-generating mechanism is commonly a loading arm or rotary actuator that rapidly twists the incident bar while the transmitted bar is fixed.

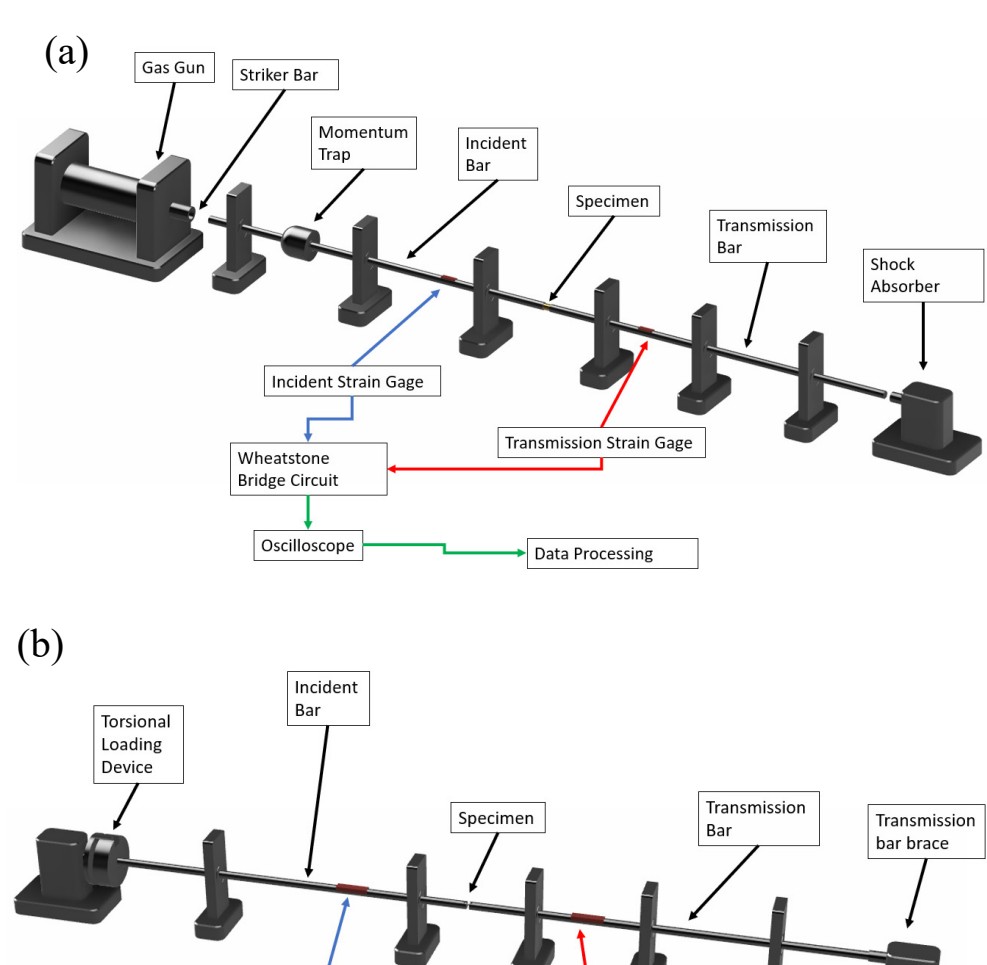

**Figure 3.** (**a**) Compressive and (**b**) torsional split Hopkinson pressure bar system.

## 3. Shear Band Formation and Propagation Kinetics

The shear region along with a fully formed ASB in a top-hat shear sample is shown in Figure 4a. Previous work showed that dynamic recovery [37], dynamic recrystallization [38], phase transformation [39], melting, and amorphization [40] can occur during the formation and propagation of an ASB [41]. Hines and Vecchio [42] proposed that progressive lattice rotations, termed PRiSM (progressive subgrain misorientation), occur during shear localization due to strain-induced recrystallization. Prior to the formation of an ASB, a larger area referred to as the shear affected zone (SAZ), or sometimes the transition layer, undergoes localized plastic shear deformation [43]. In this region, it is possible for grains to become elongated in the direction of the shear load or for grains to rotate and form high-angle grain boundaries. Large microstructural variations are observed from the outside of the SAZ to the core in which an ASB may form. The outermost sections of the SAZ have well-defined subgrains with a gradual transition to elongated subgrains near the core of the SAZ. If an ASB had formed, the subgrains at the core of the SAZ would

partition into smaller cells with highly misoriented grains due to dynamic recrystallization, though in these experiments no recrystallization was seen to occur. Distinguishing the perimeter of the SAZ and the base material is often difficult because the grains still resemble the undeformed size and shape. Electron backscatter diffraction (EBSD) scans (Figure 4b,c) can be used to get a qualitative idea of the shear deformation to identify the SAZ through the average intragranular misorientation (AIM), which is an average of the misorientation between each data point and its nearest neighbor within a grain [43]. An AIM value of 1° is often used to determine the width of a SAZ as shown in Figure 4c.

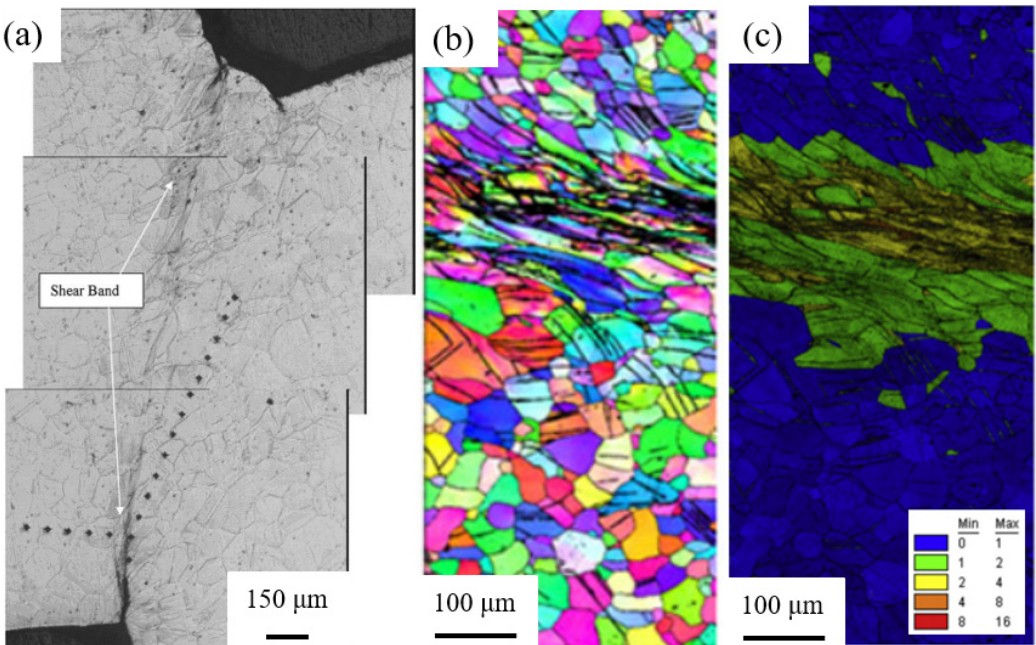

**Figure 4.** (**a**) Shear region including the shear band for a top-hat Inconel 718 sample (**b**) IPF of an ASB region and (**c**) observation of SAZ using the AIM values [43]. The scale bar for (**a**) is 150 microns and 100 microns for (**b**,**c**). Reprinted from [5,43] with permission from Elsevier.

Landau et al., studied possible indicators of ASB formation by observing the microstructure surrounding ASBs [14]. They reported that ASBs do not form from a single initiation point but intermittently within the shear region, shown in Figure 5a. TEM lamellas lifted out from regions adjacent to cracks within the ASB, marked 'A' in Figure 5a, showed dynamically recrystallized grains, Figure 5b. TEM images from lamellas lifted from regions between crack tips that formed in the ASB, marked 'B' in Figure 5a, showed heavily deformed large grains with intermittent islands of dynamically recrystallized grains, Figure 5c. The final TEM lamellas were taken 5 microns to the side of cracks within the ASB region, marked 'C' in Figure 5a. The TEM images taken from the region slightly outside the ASB still showed heavily deformed large grains but very sparse dynamically recrystallized grains, Figure 5d. Rittel et al. reported similar findings that dynamic recrystallization is a precursor to ASB formation [44]. Landau et al. experiments show that ASBs form in the shear region after sparse islands of dynamically recrystallized grains gradually evolve and coalesce into a single band of recrystallized grains [14].



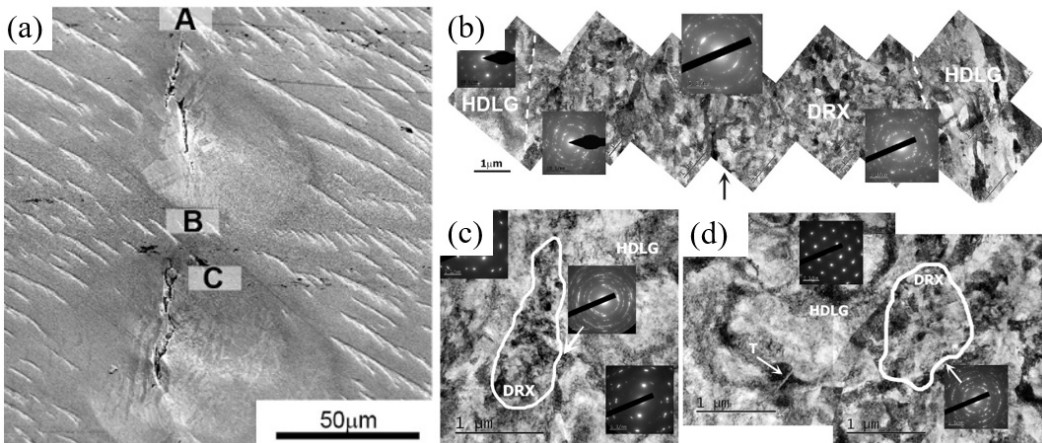

**Figure 5.** (**a**) SEM image of a shear region, TEM images from the (**b**) crack tip, (**c**) between cracks, and (**d**) 5 microns to the side of the crack. The scale bars are 50 microns for (**a**), and 1 micron for (**b**–**d**) [14].

ASBs are often seen in high-strength materials, such as steels, titanium, and aluminum alloys, susceptible to ASB formation under high strain rate loading conditions. Nickel is more resistant to shear banding and requires a much higher rate of deformation compared to other materials used in aero-engines, such as Titanium alloys, due to their high yield strengths even at elevated temperatures, low thermal softening coefficients, and low thermal conductivity, all of which affect the formation of ASBs [11]. It was observed by Hokka et al. that Inconel 625 required significantly more strain during cutting operations for ASBs to form compared to Ti-6246 [11]. The reason a higher strain is required for ASBs to form for Inconel 625 is that the thermal conductivity for Inconel 625 is twice as high as Ti 6246, therefore heat is dissipated more quickly for Inconel 625, and that Inconel 625 has a higher strain hardening exponent. This allows Inconel 625 to sustain more strain before the effects of thermal softening surpass those of strain hardening. Shear band formation during the cutting of Ti6Al4V and Inconel 718 was compared at a cutting speed of 10 m/s by Cai et al. [45]. They found that the spacing between the shear bands was approximately 80 microns for Ti64 and approximately 92 microns for Inconel 718 (Figure 6). Similar to the situation in the research by Hokka et al., Inconel 718 has almost double the thermal conductivity and nearly a 3 times higher strain hardening exponent, and thus the material was able to withstand more deformation before plastic instability could lead to the formation of a new ASB.

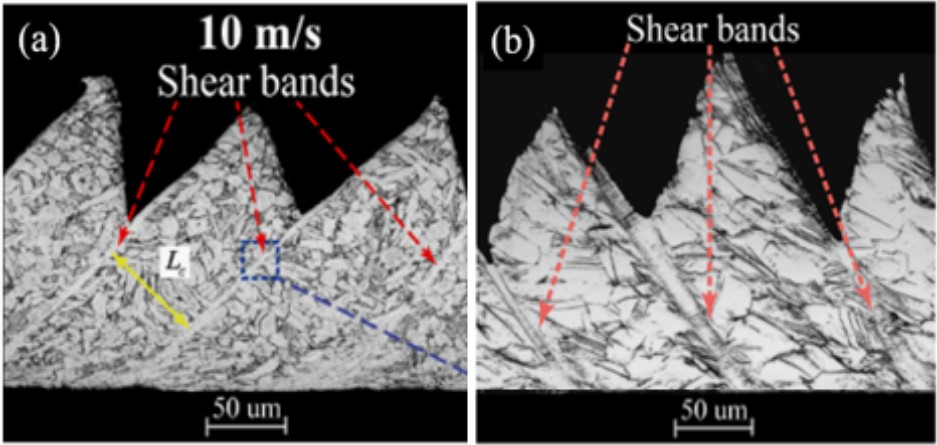

**Figure 6.** The shear band spacing during the high-speed cutting of (**a**) Ti64 is smaller than that seen in (**b**) Inconel 718 [45]. Both scale bars are 50 microns. Reprinted from [45] with permission from Elsevier.

Aging heat treatments are generally beneficial for precipitation-hardened material by increasing the yield strength through the growth of strengthening phases. However, doing so also decreases the strain required for an ASB to form due to a reduction in ductility. For Inconel 718, after aging heat treatments, when the $\gamma'$ and $\gamma''$ strengthening precipitates reach a critical size, i.e., their radii become larger than ~10 nm, a change in their shearing mechanism leads to a lower strain hardening coefficient [46]. Research by Song et al. [18] conducted dynamic shear tests using a top hat sample geometry and stopper rings of various thicknesses on solution treated (Figure 7a) and aged (Figure 7b) Inconel 718. While the aged material did show a higher shear yield strength than the solution-treated material, the critical strain for an ASB to form was lower for the aged material. Under the same testing conditions, the aged material formed ASBs between strains of 2.4 and 3.2 while the solution-treated samples required a strain of 4.5 for an ASB to form. A summary table of dynamic shear experiments on nickel-based superalloys is shown in Table 1.

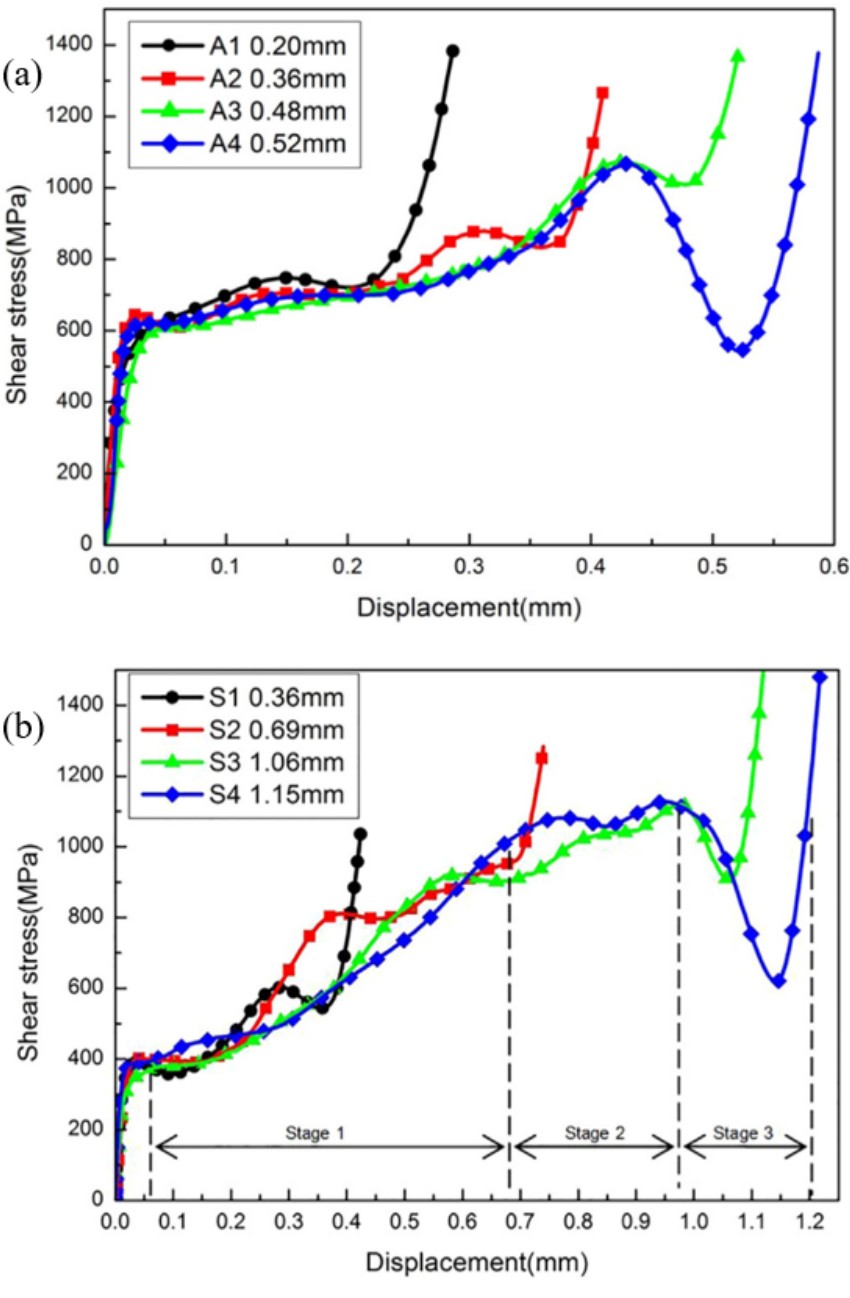

**Figure 7.** Shear stress vs. strain graphs for (**a**) solution treated and (**b**) aged samples [18]. Reprinted from [18] with permission from Elsevier.

**Table 1.** Summary of dynamic shear testing under various heat treatments and testing strain rates [5,18,47–49].

| Year | Author | Material | Tested Strain Rate (s$^{-1}$) | Heat Treatment | Width of ASBs (Microns) |
|---|---|---|---|---|---|
| 2003 | Clos | Steel Inconel 718 | Shear: $10^6$–$10^7$ | - | 4–20 2–4 |
| 2009 | Demange | Inconel 718 | Shear: $5 \times 10^4$ | Annealed Precipitation hardened | 65.8 33.6 |
| 2016 | Johansson | Inconel 718 | Global: approximately 1500 | Precipitation hardened | Top hat:4–5 Cutting: 4–5 |
| 2018 | Song | Inconel 718 | Shear: $8 \times 10^4$ | Solution treated Aged | 10–13 10 |
| 2020 | Colliander | Inconel 718 | - | Precipitation hardened | 7 |

## 4. Microstructures in an ASB

In addition to dynamic recrystallization, the microstructure within the ASB and SAZ experience significant changes in terms of precipitate morphology. In particular, the dissolution of strengthening phases in the ASB region has been observed, though the mechanism behind their dissolution requires further research [49]. Atom probe tomography (APT) specimens were taken from undeformed and deformed Inconel 718 material. An optical image of an ASB in the deformed sample is shown in Figure 8a. This material shows clusters of aluminum, titanium, and niobium which indicates the presence of $\gamma'$ and $\gamma''$ strengthening phases, Figure 8b,d. However, APT specimens from an identified ASB in deformed cylindrical top hat samples showed no clustering which is evidence of an absence of strengthening phases, Figure 8c. Estimations of the temperatures in the ASB obtained by numerical integration of the thermal balance equation (assuming adiabatic conditions and neglecting thermo-elastic effects) were between 864 °C and 1118 °C. These values are in the range of solvus temperatures for the $\gamma'$ and $\gamma''$ strengthening phases, 850 °C and 900 °C respectively. The surrounding material does not increase in temperature; thus, it effectively quenches the ASB and limits the time the ASB is above the solvus temperature of $\gamma'$ to less than one µs, which prevents the nucleation of precipitates.

Inconel 718 has been shown to have a higher resistance to ASB formation compared to low-carbon steel [47]. The microstructure in Inconel 718 was observed to be more stable after shear localization compared to low-carbon steel by Clos et al. [47]. In their study, microstructural observations of flat top-hat dynamic shear samples of low-carbon steel showed a nearly linear growth of shear band thickness from 4 microns to 20 microns after shear localization. Similar Inconel 718 samples showed a constant band width, approximately 2–4 microns, but no increase in size after shear localization. The critical strain values for the formation of ASBs were 2.2 for the steel and 1.7 for Inconel 718. Following the onset of unstable shear localization, the stress was reduced, and the remaining deformation was confined to a narrow zone. Temperature measurements using InSb-detectors and InGaAs-detectors on the flat top hat samples measured temperatures of 500 °C after initial shear localization and 800 °C in the post-localization phase of the experiments with a heating rate on the order of $10^7$ K/s.

Johansson et al. [48] compared the formation process of ASB between Inconel 718 sheared at a high strain rate in a top hat sample geometry and under a high-speed cutting operation. The top hat samples and the samples mechanically cut under high speeds showed a similar appearance on the microstructural level with both having visible shear bands approximately 4–5 microns wide. The Inconel samples showed no evidence of a transition

zone between the ASB and the base material. Grains near ASB were observed to be in the process of subdivision by shearing. Evidence of dynamic recrystallization was not observed using EBSD due to low confidence indexes in the ASB region (Figure 9b), however, band contrasts maps taken from a transmission electron microscopy (TEM) foil removed from the internal section of the ASB outlined in red (Figure 9a) showed ultra-fine equiaxed grains, Figure 9c, with a size between 50 and 300 nm. This further reinforced that the region is an ASB because it has undergone recrystallization and is not simply a SAZ.

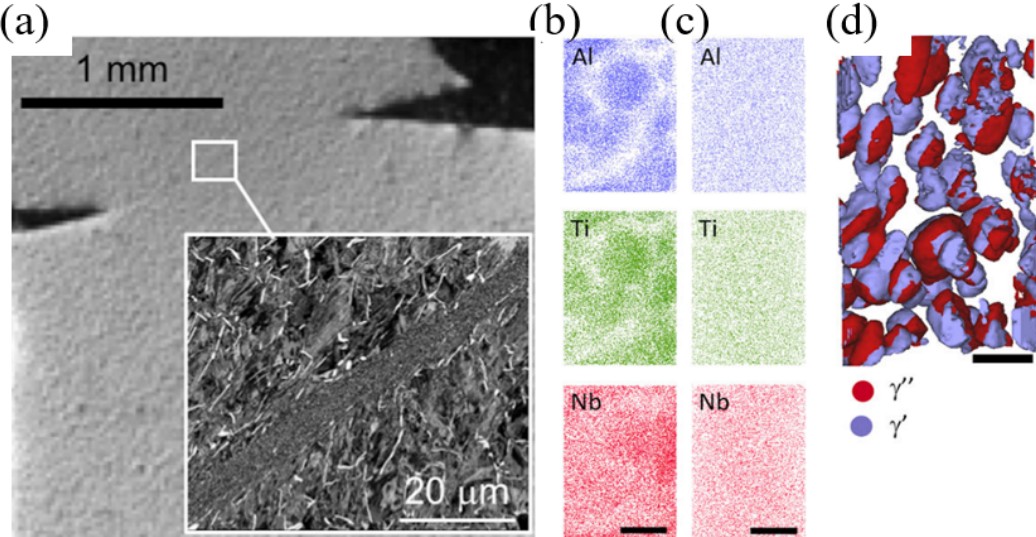

**Figure 8.** (**a**) Image of the ASB from which APT samples were lifted out from. Chemical distribution of precipitate elements from (**b**) undeformed bulk material, (**c**) ASB region, and (**d**) iso-surface reconstructions of precipitates from the undeformed bulk material. A 1 mm scale bar is used for (**a**) and (**b**–**d**) scale bars are 20 nm. Reprinted form [49] with permission from Taylor and Francis.

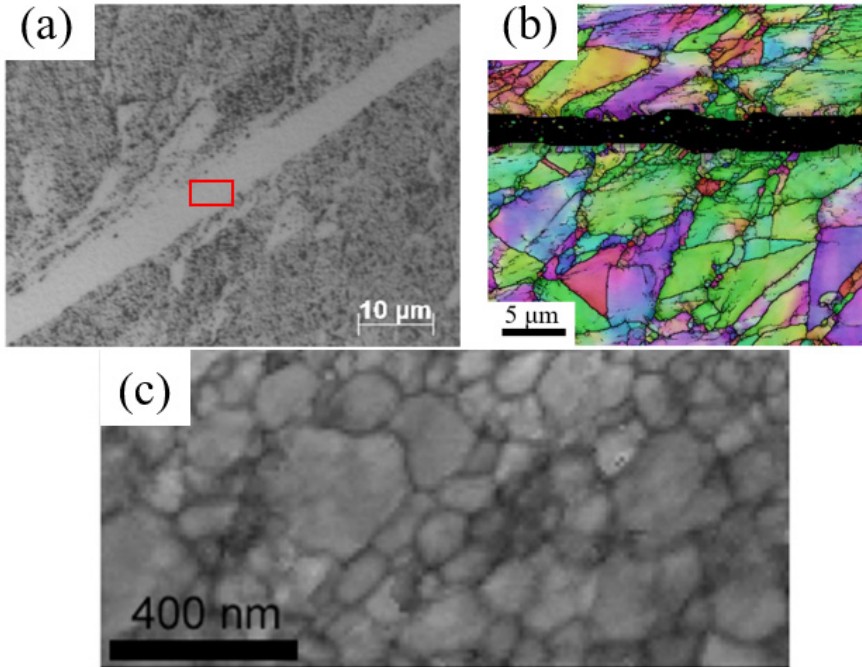

**Figure 9.** (**a**) Etched OM image, (**b**) IPF of the shear band, and (**c**) Band Contrast image showing recrystallized grains from a TEM foil lifted out from the ASB region (red box) [48]. The scale bar for (**a**) is 10 microns, (**b**) is 5 microns, and (**c**) is 400 nm. Reprinted from [48] with permission from Elsevier.

Severe plastic deformation in the shear region can lead to work hardening by an increase in the dislocation density, Figure 10a. Since the deformation is localized primarily along the shear plane, the surrounding microstructure is mostly unaffected, but sometimes shows slight grain elongation and rotation along the direction of the shear load. Using this information, ASBs can be identified using EBSD as the grains will be slightly misoriented. If the material in the ASB has undergone dynamic recrystallization, the grains will be very small and require a very low step size to accurately measure the recrystallized grain's size. Identification of the approximate size of the ASB is possible using EBSD even though the step size is not small enough to record the size of the recrystallized grains because the ASB will consist of a band of points with a very low confidence index [18] (Figure 10b).

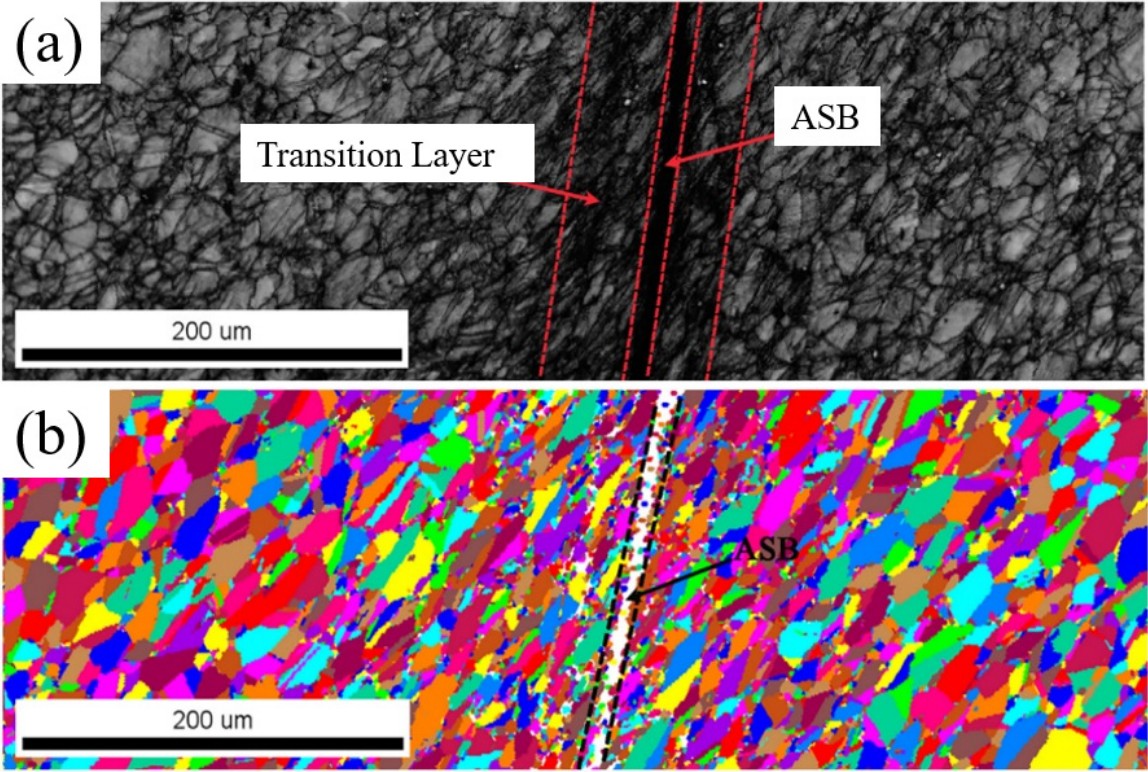

**Figure 10.** (**a**) Band contrast and (**b**) IPF maps of the ASB and surrounding material [18]. Both scale bars are 200 microns. Reprinted from [18] with permission from Elsevier.

An image of the microstructure in and around an ASB is shown in Figure 11a. Measurements of the hardness values in the SAZ and ASB regions can be used as qualitative estimations of their width. It was reported previously that the hardness values in the SAZ increase due to an increase in dislocation density [5,18,43,50–52]. Inconel 718 specifically contains both FCC Ni which facilitates slipping and BCC Fe which favors the formation of mechanical twins [5]. The microhardness measurements taken from the SAZ region in Inconel 718 which was dynamically deformed under shear loads are shown in Figure 11b with a maximum value at the center of the ASB. The indents of the hardness measurements are shown in Figure 11a and the dashed red line traces the path of the ASB. Hardness values are reported as averages of each column with the maximum and minimum values from each column shown by the error bars. Since material in the ASB region is recrystallized and exhibits a smaller average grain, the material hardens due to the Hall-Petch relationship which describes the inverse relationship between yield strength and grain size.

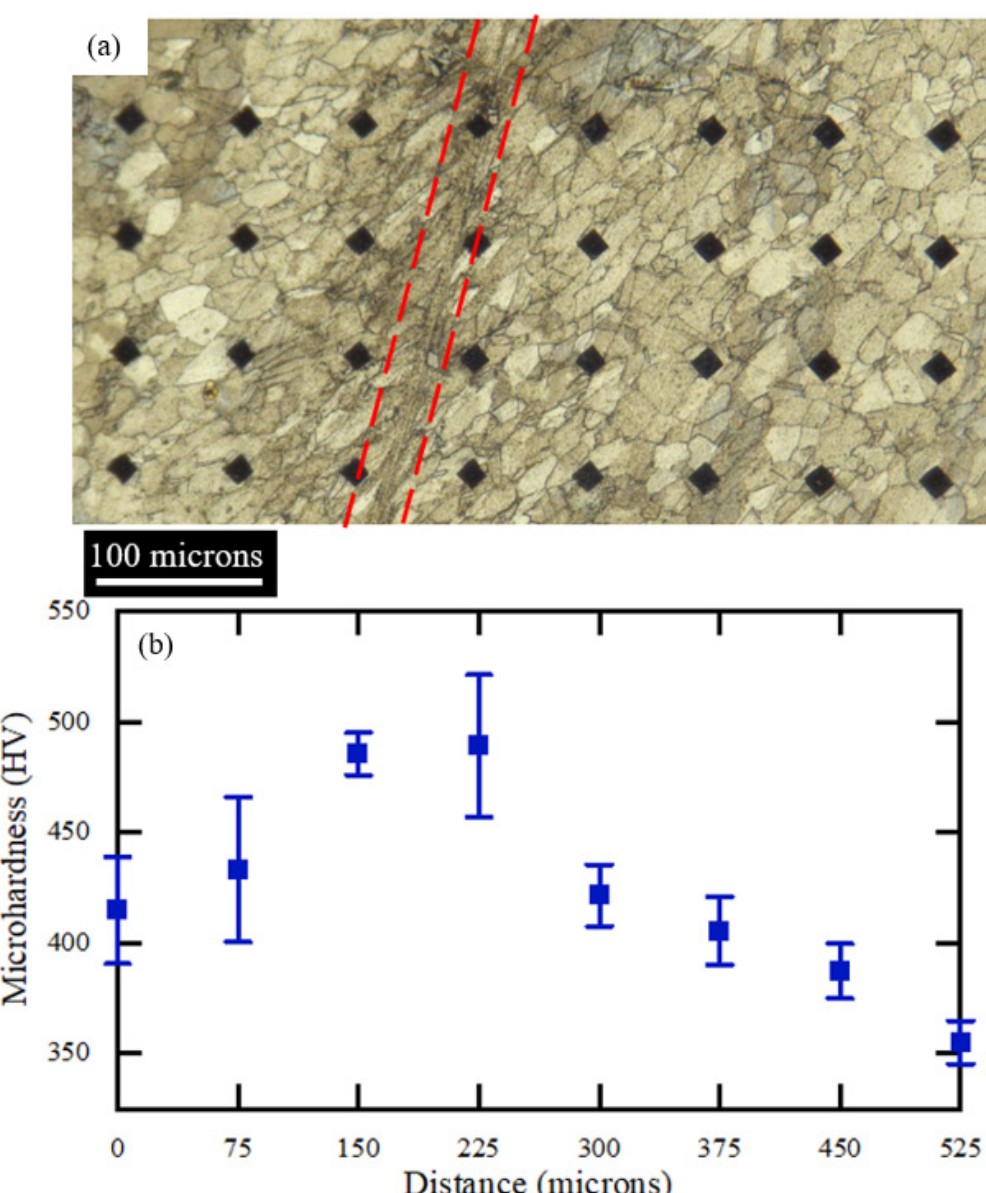

**Figure 11.** (**a**) OM image showing the SAZ region of the shear band and (**b**) micro-hardness measurements perpendicular to the ASB. The scale bar in (**a**) is 30 microns.

## 5. Conclusions

ASBs are rare and unpredictable dynamic failure mechanisms that occur at very high strain rates. ASBs form because the heat generated by rapid plastic deformation does not have sufficient time to dissipate to the surrounding material and thus thermally softens the material within the shear band, leading to localized shear failure. In order for an ASB to form both a critical strain and critical strain rate must be reached, though given that this is a probabilistic process it is not guaranteed to occur even if the conditions are met. The critical strain value for aged Inconel 718 has been recorded to be around 2.2 and 3.2 while the solution-treated material required strain values of 4.5 in samples tested at a shear strain rate of $8 \times 10^4$ s$^{-1}$. The higher resistance to shear localization in the solution-treated sample was attributed to the presence of $\gamma'$ and $\gamma''$ phases which pin grain boundaries thereby assisting with work hardening. Furthermore, AM Inconel 718 has a lower critical strain value compared to cast material, which has been attributed to the voids and defects inherent in the AM process. Shear band widths vary widely for Nickel-based superalloys ranging between 2 microns and 65 microns. This large variation is due to the effect heat

treatments have on ASB formation. The band width for Inconel 718 is 65.8 microns after annealing and 33.6 after aging heat treatments. Unlike other materials that commonly experience ASBs, no new phases have been recorded to form in the shear bands of Inconel. Few investigations into the formation of ASBs in AM materials have been conducted on nickel-based superalloys and is limited to the high-speed cutting of AM Inconel 625. Future work on this topic could include comparing the shear localization behavior of AM and TM materials using the same sample geometry, such as the top hat, under the same shear strain rates. This will provide a better understanding of the ASB formation mechanism in AM materials and help with developing more comprehensive and versatile models. In conclusion, ASBs are a probabilistic failure phenomenon that occurs under a high strain rate in Nickel-based superalloys, and understanding their formation mechanisms in nickel-based superalloys is key to the study of high-strain damage mechanisms in aerospace-grade materials.

Notable findings from the works referenced in this paper include the following:

1. ASB bandwidths vary between 2 microns and 65.8 microns for Nickel-based superalloys.
2. Aging heat treatments on nickel-based superalloys decrease the strain required for an ASB to form from 4.5 to between 2.2 and 3.2 and nearly halves the band widths of the ASB.
3. No new phases precipitate during ASB formation in Inconel 718 however $\gamma'$ and $\gamma''$ strengthening phases are reported to dissolve.

**Author Contributions:** Supervision, defining the project, funding acquisition, K.D. and P.G.A.; writing and original draft preparation, R.A.R.; review and editing, K.D., P.G.A. and A.N.P. All authors have read and agreed to the published version of the manuscript.

**Funding:** This research received no external funding.

**Institutional Review Board Statement:** Not applicable.

**Informed Consent Statement:** Not applicable.

**Data Availability Statement:** Not applicable.

**Acknowledgments:** The authors would like to thank the support of The U.S. Air Force Research Lab Summer Faculty Fellowship Program.

**Conflicts of Interest:** The authors declare no conflict of interest.

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
