# Peer review of "Adiabatic Shear Banding in Nickel and Nickel-Based Superalloys: A Review"

_metals, doi:10.3390/met12111879_

Round 1

Reviewer 1 Report

This review paper describes adiabatic shear banding in nickel and nickel based alloy. I think it might be suitable for publication in Materials once the authors tackle the following questions.

1.why the imperfection in AM can easily trigger ABS comparing with traditional manufacturing process ? Also, what is the origin of different morphologies of ABS in different manufacturing process?

2.it seems like the authors take much effort to introduce the basic testing methods. I think the main concern in this paper should be the scientific/foundational part of ABS instead of testing methods. So I suggest the authors should consolidate part2.

3.As for SAZ and ABS, what is the most important physical feature that people can distinguish, especially for the materials without recrystallization ?

4.the authors should further explain why the formation of ABS is different between aged material and solution treated material in physical and metallurgical aspect. 

5.it is too vague in the description of part 4( microstructural in an ASB). the authors should make more comprehensive literature review in part 3 and par4 so as to give the audience a complete picture of ASB.

Author Response

Dear Editor,

We would like to thank the reviewers again for their constructive comments. The manuscript was revised as suggested and specific responses to the reviewers’ questions are provided below in BLUE.

Reviewer 1:

  1. Why the imperfections in AM can easily trigger ASB comparing with TM process? Also, what is the origin of different morphologies of ASB in different manufacturing process?

Thank you for bringing these questions to our attention. An explanation of how imperfections in AM can easily trigger ASBs was added (lines 65-69) and is copied here for the reviewer’s convenience:Material that has been additively manufactured (AM) commonly contains imperfections, such as microvoids, which create stress concentrations and allow ASBs to form at lower strains compared to traditionally manufactured (TM) material.” Likewise, sentences were added to explain why different ASB morphologies can form between AM and TM material (lines 69-71).

  1. It seems like the authors take much effort to introduce the basic testing methods. I think the main concern in this paper should be the scientific/foundational part of ASB instead of testing methods. So I suggest the authors should consolidate part 2.

We certainly agree with the reviewer that the testing methodology section was too lengthy. To keep the focus of this paper on the understanding of ASBs in nickel-based alloys we have consolidated part 2 (testing methodology).

  1. As for SAZ and ASB, what is the most important physical feature that people can distinguish, especially for the materials without recrystallization?

Thank you for your comment. In the class of material discussed in this paper, nickel-based superalloys, the most important physical feature to identify an ASB is recrystallization. The most important feature to identify the SAZ is an increase in the AIM (average intragranular misorientation) value. To clarify this with the audience, an explanation was added to the introduction (lines 90-91). Additionally, lines 188-191 were edited to state the SAZ could be identified by measuring the AIM values and the a related reference was added to the end of the sentence in line 191 (see Cerreta, E.K.; Bingert, J.F.; Gray, G.T.; Trujillo, C.P.; Lopez, M.F.; Bronkhorst, C.A.; Hansen, B.L. Microstructural Examination of Quasi-Static and Dynamic Shear in High-Purity Iron. Int J Plast 2013, 40, 23–38, doi:10.1016/j.ijplas.2012.06.005.).

  1. The authors should further explain why the formation of ASB is different between aged material and solution treated material in physical and metallurgical aspect.

The authors thank the reviewer for bringing this to our attention. We have added information to explain that the formation of ASBs is different between the aged and solution treated material due to precipitate hardening effects (lines 247-250, Reference 46).

  1. It is too vague in the description of part 4 (microstructure in an ASB). The authors should make more comprehensive literature review in part 3 and part 4 as to give the audience a complete picture of ASB.

The authors would like to thank the reviewer for this comment. To give further description of the ASB, additional OM images of ASBs were added to Figure 4 and Figure 8. We also added a new paragraph to part 3 describing how the microstructure evolved leading up to ASB formation (lines 201-214). Additionally, a new Figure 5 was added to provide visual aid for the new paragraph.

Reviewer 2 Report

This paper studied to Adiabatic shear banding in nickel and nickel-based superal-2 loys: a review. The article is, in general, well written but there are some issues that authors should consider to revise in order to improve its quality. Some comments were done in this way:

·         Abstract should be expanded sentences related to the results. The results of the study should be given as numerical percentages.

·         The introduction is short, it should be expanded. Only the most relevant and up-to-date articles on the study should be given. The paper should be also supported by a literature search including relevant and recent papers. The following recent articles related to optimization may be cited.

https://doi.org/10.1007/s00170-010-3012-9

·         Give the split Hopkinson pressure bar system test parameters with a table.

·         Revise Figure 7-9, make it clearer and higher resolution.

·         The article should be edited completely according to the journal writing guide.

·         Throughout the article, the words table and figure should start with capital letters (Table, Figure).

·         Standards of test samples and deniers should be given.

·         Fractions should be given with dots throughout the article, including figures and tables.

·         Conclusions should be written in more detail adding numeric data. Also, give the results in items.

Author Response

Dear Editor,

We would like to thank the reviewers again for their constructive comments. The manuscript was revised as suggested and specific responses to the reviewers’ questions are provided below in BLUE.

  1. Abstract should be expanded sentences related to the results. The results of the study should be given as numerical percentages.

Thank you. We have added sentences to the end of the abstract to reference noteworthy results (lines 24-28).

  1. The introduction is short, it should be expanded. Only the most relevant and up-to-date articles on the study should be given. The paper should be also supported by a literature search relevant and recent papers. The following recent articles related to optimization may be cited: https://doi.org/10.1007/s00170-010-3012-9.

We agree with the reviewer in that the introduction would be improved by including up-to-date articles and thank them for suggesting the article on optimization. We have added another paragraph to the end of the introduction (lines 96-101) and included the recommended reference (reference 22) along with a couple more relevant articles (references 20 and 21). Additionally references 2, 18, and 19 were added to the introduction.

  1. Give the split Hopkinson pressure bar system test parameters with a table.

Thank you. A table listing the material, strain rates, heat treatments, and width of ASBs for several relevant papers was added (line 260).

  1. Revise figure 7-9, make it clearer and higher resolution.

Thank you for calling this to our attention. We have revised the pictures by increasing the resolution and making them slightly larger. We have also edited the caption on Figure 9 (now it is Figure 10) so it is easier to read.

  1. The articles should be edited completely according to the journal writing guide.

The authors would like to thank the reviewer for this comment. We have revised the entire manuscript to ensure that our writing upholds the journal writing guides instructions.

  1. Throughout the article, the words table and figure should start with capital letters.

Thank you for catching our mistake, we have corrected our manuscript so that every time “table” or “figure” is used it starts with a capital letter.

  1. Standards of test samples and deniers should be given.

To the best of our knowledge, there are no standards for testing using an SHPB.

  1. Fractions should be given with dots throughout the article, including figures and tables.

Thank you for calling this to our attention, all figures and tables have been written in the following format (Figure #., and Table #.)

  1. Conclusion should be written in more detail adding numeric data. Also give the results in items.

Thank you. We have added more numerical data to the conclusion. The information was added on lines 379-380 and lines 386-387. A table listing the material, strain rates, heat treatments, and width of ASBs for several relevant papers was added to line 260. Additionally, several notable findings are reported at the end of the conclusion (lines 399-406).

Reviewer 3 Report

The paper entitled " Adiabatic shear banding in nickel and nickel-based superalloys: a review" presents an extensive literature review of the shear banding in nickel-based superalloys. From my point of view, the article is of great interest but more analysis could be done to improve overall quality:

·        A graphical abstract would add interest to catch the eye

·        It would be interesting to give more visibility to shear band analysis in alternative processes to machining (welding, additive manufacturing...).

·        Add some summary table of test bench equipment for nickel based material testing.

·        I think it would be interesting to include some new works on the field:

o   https://doi.org/10.1007/s11340-022-00887-x

o   https://doi.org/10.3390/cryst10080689iabatic Shear Localization Frontiers and Advances, 2nd ed.; Elsevier: London, UK, 2012.

·        Rewrite the conclusions by drawing conclusions of a review paper with a summary of the activities performed and topics covered.

·        In relation to the previous review of guidelines in Future Lines of those things that can be worked on further, as well as the gaps that need to be filled with future research.

These comments are intended to contribute to the improvement of the quality of the paper presented.

Author Response

Dear Editor,

We would like to thank the reviewers again for their constructive comments. The manuscript was revised as suggested and specific responses to the reviewers’ questions are provided below in BLUE.

  1. Graphical abstract would add interest to catch the eye.

Thank you for the idea. The graphical abstract will not upload to this box. I added the graphical abstract at the end of the manuscipt (after the references) for your review.

  1. It would be interesting to give more visibility to shear band analysis in alternative processes to machining (welding, additive manufacturing…)

To the best of our knowledge, ASBs do not form during welding or additive manufacturing processes.

  1. Add some summary table of test bench equipment for nickel based material testing.

Thank you. A table listing the material, strain rates, heat treatments, and width of ASBs for several relevant papers was added (line 260).

  1. I think it would be interesting to include some new works on the field:

https://doi.org/10.1007/s11340-022-00887-x

We appreciate the suggestion for new works on the torsional testing of Inconel 718 and the mentioned work has been cited in the testing methodology section on line 106 (please, see reference 26).

https://doi.org/10.3390/cryst10080689

We appreciate the reviewer’s suggestion for additional references for new methods of additively manufacturing Inconel 718, the mentioned work was cited (lines 66, please, see reference 15)

  1. rewrite the conclusions by drawing conclusions of a review paper with a summary of the activities performed and topics covered.

The authors thank the reviewer for this recommendation. A discussion was added in lines 386-387 and 389-395. A summary of several notable findings is also included at the end of the conclusion, lines 399-406.

  1. In relation to the previous review of guidelines in Future Lines of those things that can be worked on further, as well as the gaps that need to be filled with future research

Thank you. We have added suggestions for future works based upon gaps in the current literature to our conclusion (lines 390-395).
